# Structure and Nanomechanics of PPTA-CNT Composite Fiber: A Molecular Dynamics Study

**DOI:** 10.3390/nano12183136

**Published:** 2022-09-09

**Authors:** Tong Li, Zebei Mao, Juan Du, Zhuoyu Song

**Affiliations:** Department of Engineering Mechanics, Dalian University of Technology, Dalian 116024, China

**Keywords:** CNT, Poly(ρ-phenylene terephthalamide), nanomechanics, molecular dynamics

## Abstract

Poly phenylene terephthalamide (PPTA) fiber has both high mechanical properties and low thermal conductivities, making it ideal for the design of thermal protection material in hypersonic vehicles. In this paper, the impact of CNT additions on the nanostructure and mechanical performances of PPTA fibers is investigated by coarse-grained molecular dynamics (CGMD) simulation. It can be found that CNT addition performs as the skeleton of PPTA polymer and induces a higher degree of alignment of polymers under shear deformation during the fabrication process. Both strength and Young’s modulus of the PPTA fiber can be improved by the addition of CNTs. The interaction between CNTs and PPTA polymer in PPTA fiber is important to further improve the efficiency of force transfer and mechanical performance of PPTA-CNT composite fibers.

## 1. Introduction

With the development of various lightweight equipment, fiber-reinforced plastics (FRP) attracted more and more attention because of their superior mechanical, thermal, electrical, and ballistic behaviors [1]. Poly(ρ-phenylene terephthalamide) (PPTA) fiber is one of the most widely used synthetic fibers as the reinforcement in these FRPs due to its high toughness and abrasive characteristics [2]. The high strength and low thermal conductivity make PPTA fibers an ideal candidate for thermal protection materials in aircraft and space stations [3,4,5]. However, with the development of hypersonic vehicles, both the extremely high working temperature and the load-bearing ratio of the vehicle structures call for further improvement of the mechanical properties of PPTA fibers.

PPTA fibers have 1,4 para-substituted aromatic rings (providing thermal stability), and the rationality of its outstanding mechanical properties can be attributed to the well-aligned crystal structures, which were investigated by the bottom-up multiscale modeling technique [6]. The basic mechanical properties of the Kevlar fiber are affected by a few structural factors, such as skin thickness, width, and misorientation angle of microscale pleats. A few efforts were devoted to further improving the mechanical performances of PPTA-reinforced composites. The surfaces of PPTA fibers were grafted with ethylene glycol diglycidyl ether to introduce epoxy groups on the surface of PPTA fibers, and the interfacial strength between PPTA fiber and rubber matrix can be improved by 67.5% [7]. Kanbargi and Lesser altered the fiber surface morphology to allow for penetration of reactive monomers into the fiber subsurface, which doubled the interfacial strength between PPTA fibers and rubber materials [8]. The surface and interface modification of PPTA fibers as reinforcement for polymer composites were well reviewed in the literature [9]. These studies focus on the treatment of fiber/matrix interaction that can improve the overall performance of composite products.

There is limited research on the modification of PPTA fibers to improve their mechanical properties of PPTA fibers. Carbon nanotube (CNT) is a commonly used carbon nanomaterial in the last three decades to improve the mechanical performance of various materials. The interfacial adhesion between the rubber matrix and PPTA fibers can also be adjusted by grafting CNTs onto PPTA fibers [10]. PPTA oligomers were grafted onto carbon nanotubes (CNTs) and the tensile modulus and strength were improved by 15~20% compared to untreated PPTA fibers [11]. However, this outcome is still below the theoretical limit of mechanical improvement of PPTA fiber by CNTs predicted by molecular dynamics (MD) methods [12]. The reinforcing efficiency of CNTs is limited by the low intrinsic shear strength of these highly anisotropic polymer fibers, which limits effective stress transfer and nanotube reinforcement [13].

Normally, PPTA fibers are fabricated by the dry-jet wet spinning method [14], in which process the nano-additions, such as CNTs, could significantly change the nanoscale structures of the PPTA products, therefore, affecting the mechanical performance of the fiber products. Young et al. fabricated PPTA-CNT composite fiber by dry-jet wet spinning method and found that the orientation of polymer chains in the composite is sensitive to the nano-additions [15], which can improve the mechanical performances of PPTA fibers [16]. The orientation of polymers is also sensitive to the speed of fiber take-up during fabrication [17], but the underlying molecular mechanism of how CNT additions mediate the orientation of PPTA polymers still needs further investigation.

In this paper, MD modeling of the dry-jet wet spinning process of PPTA fibers was conducted, which also considered the addition of CNTs under different Weissenberg numbers. The polymer orientation and mechanical performance of CNT-reinforced PPTA fibers were studied to understand the impact of CNTs on the mechanical performance of PPTA-CNT composite fiber.

## 2. Methods

The molecular modeling of PPTA fabrication with the sulfuric acid solution is conducted by using the coarse-grained molecular dynamics (CG-MD) method [18]. The force field parameters of the PPTA system are obtained from the work by Mogurampelly et al., including the CG force field parameters of the PPTA-H_2_SO_4_ solution system [19] and the basic bead-simplification model is provided in Figure 1. The 9-6 Lennard–Jones, bond, angle, and dihedral interaction parameters are all provided in Appendix A.

The all-atom model of the PPTA system was developed on Material Studio 7.0 (Accelrys, San Diego, CA, USA) and was converted to a CG model by using Gromacs 2018 [20]. Each polymer chain has 8 repeating units, as provided in Figure 1; a total of 2500 PPTA molecular chains and 200,000 H_2_SO_4_ molecules are considered in the solution system. In total, 50 armchair single-walled CNTs with a length of 10 nm were added to the PPTA system using the CGMD force field developed by Chou et al. [21]. The insertion position and angle of CNTs are random. This material content will result in a 9.5% weight fraction of CNTs in the final PPTA fiber. The molecular simulation was all carried out on a workstation (CPU: Intel Xeon Gold 6133, GPU: NVIDIA TITAN RTX) using parallel calculation. 

The initial size of the simulation system is 90 × 90 × 90 nm with periodicity in all directions. An energy minimization simulation was conducted with a tolerance of 10 kJ/mol·nm, followed by 1 ns constant pressure and constant-temperature ensemble (NPT) simulation (1 bar and a temperature of 300 K). Then, 10 ns canonical ensemble (NVT) simulation were carried out at 300 K after the NPT simulation. The shear deformation of the PPTA system is simulated with all the other dimensions unchanged, and the shear deformation is driven by the deformation of the box. The shear rate is 0.35 ns^−1^ and the Weissenberg number, which represents the degree of shear deformation, is 20 in this work. After the shear deformation modeling, NVT and NPT simulations with both 10 ns periods were conducted to model the process of fiber solidification and refinement. It should be noted that the sulfuric acid (H_2_SO_4_) molecules were removed from the system as diffusion of sulfuric acid molecules happens during the last step of the fabrication in water. X-ray scattering intensity calculation method [22] was employed to characterize the nanostructure of PPTA-CNT composite fiber. The molecular system of PPTA-CNT composite fiber after refinement is illustrated in Figure 2.

## 3. Results and Discussion

In previous studies, the ability of CNTs to increase the mechanical properties of PPTA fibers depends on the final molecular configuration of PPTA-CNT composite fiber [6,12]. It is necessary to study the effect of CNTs on the nanostructure of PPTA-CNT composite fiber during the dry-jet wet spinning process.

### 3.1. Structural Changes by CNTs

Figure 3 and Figure 4 show the nanostructure formation process of the PPTA system containing CNTs with Wi ranging from 0 to 20, respectively. It can be recognized that CNTs have a significant orientation when the system is subjected to shear deformation. During the solidification process, CNTs become the skeleton of the PPTA molecular chain aggregation, affecting the spatial structure of the PPTA system. Then a compact system is formed after fiber refinement. Different from the PPTA system without CNT, the axial size of the PPTA system with a Wi of 20 after refinement is 25.5, while the axial size of the system without CNT is 26.3. This suggests that PPTA without CNTs is more likely to become elongated during the fabrication process.

Another difference from the PPTA system without CNT is that the spatial structure of the PPTA system with CNTs is significantly affected by the orientation of the CNTs during solidification. As shown in Figure 5a, when the Wi is 0, the direction of CNTs is disordered, and the spatial network structure formed by PPTA molecular chains and CNTs is similar to the system without CNTs. As shown in Figure 5b, when the Wi is 20, the orientation of the CNTs is very directional, and the PPTA molecular chains aggregate around the CNTs to form layered microfibers, which are different from the system without CNTs. It can be concluded that the CNTs added to the PPTA system play as a skeleton in the molecular system, which connects the PPTA molecular chains. After shear deformation, the CNT skeleton has an obvious orientation, which drags the molecular chains in the same deformation mode. Therefore, compared to the system without CNTs, the molecular chains in the PPTA system with CNTs are more aggregated, forming a nanostructure with fewer voids.

### 3.2. Mechanical Properties Changed by CNTs

From a literature review, it can be concluded that the overall mechanical performance of PPTA benefits from the addition of CNTs, a 10% addition of CNTs could improve Young’s modulus and strength of PVA fibers by 10~15% [16]. We conducted mechanical deformation modeling of the PPTA-CNT composite fiber and the stress–strain curve of the PPTA-CNT composite fiber is summarized according to the CNT content and shear deformation (Wi number). 

The stress–strain curves of PPTA and PPTA-CNT composite fiber are summarized in Figure 6. The stress–strain curve before 0.05 strain is used to obtain Young’s modulus of the material. The strength and Young’s modulus of these fibers are provided in Table 1.

Young’s modulus of PPTA fibers is sensitive to shear deformation. For PPTA and PPTA-CNT fibers, Young’s modulus can be improved by 46.6% and 49.9% by shear deformation of Wi = 20, correspondingly. Similarly, the strength of materials can also be mediated by the shear deformation, and the degree of improvement for these two systems is 6.6% and 9.0%. The improvements validated the conclusion in the literature that a higher draw ratio of fibers in the fabrication process can sensitively mediate the mechanical performance of these polymer fibers. This is because a shear deformation can change the state of aggregation and orientation of polymer chains. A larger shear deformation can induce the alignment of polymer chains in the PPTA fiber, which is positive for mechanical improvement.

Compared to the PPTA fibers, the reinforcement of CNTs during the dry-jet wet spinning process can also significantly improve the mechanical properties of PPTA fibers. Take the fibers under the Wi = 20 condition as an example; the strength and Young’s modulus are improved by 14.52% and 17.9% with a 9.5% weight fraction of CNT addition. The molecular configuration of PPTA-CNT fibers after mechanical deformation simulation is provided in Figure 7 to study the mechanisms of force transfer in the composite fiber.

It can be found that the voids in the material during stretching happen near the aggregation of CNTs; however, the surface of CNT material is smooth, limiting the ability of force transfer between polymer matrix and CNT additions. Previous studies showed that the surface modification of CNTs can enhance the interfacial shearing strength between CNTs and polymers [8,11], improving the overall mechanical performances of CNT-reinforced polymers. The future development of synthetic fibers with outstanding mechanical behaviors calls for both improved interface strength and induction of polymer alignment in the PPTA-CNT composite fibers.

## 4. Conclusions

To understand the molecular mechanisms of how CNT additions change the nanostructure and mechanical performance of PPTA-CNT composite fibers, coarse-grained molecular dynamics simulations were carried out to simulate the molecular configuration of the PPTA-CNT system during fabrication and mechanical deformation. The following conclusion can be obtained:(1)CNT addition performs as the skeleton of PPTA polymer and induces a higher degree of alignment of polymers under shear deformation during the fabrication process. The molecular chains in the PPTA system with CNTs are more aggregated with fewer voids.(2)Both strength and Young’s modulus of the PPTA fiber can be improved by the addition of CNTs; however, the damage still initiates near the CNTs. The improvement of interfacial shearing strength is important to improve the efficiency of force transfer between CNTs and PPTA polymer.

## Figures and Tables

**Figure 1 nanomaterials-12-03136-f001:**
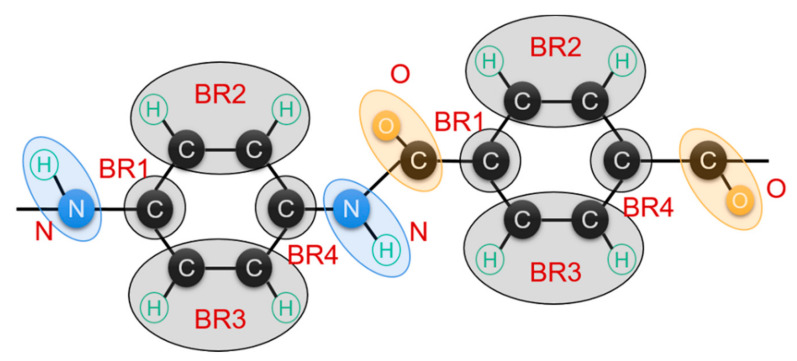
CG simplification model of PPTA monomer based on the all-atom model.

**Figure 2 nanomaterials-12-03136-f002:**
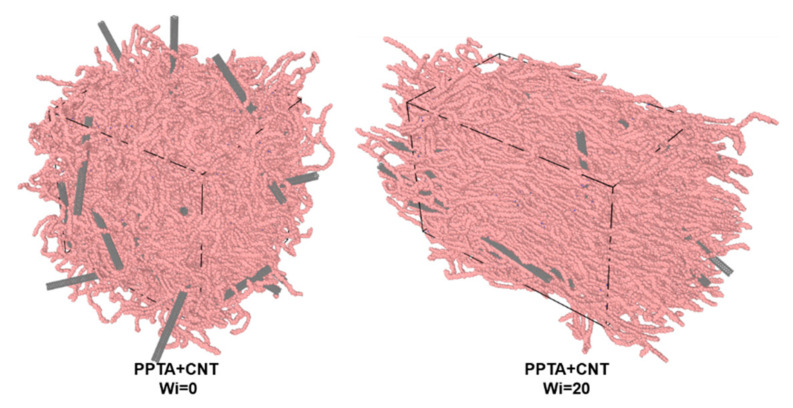
Molecular system for mechanical simulation of PPTA + CNT composite fiber.

**Figure 3 nanomaterials-12-03136-f003:**
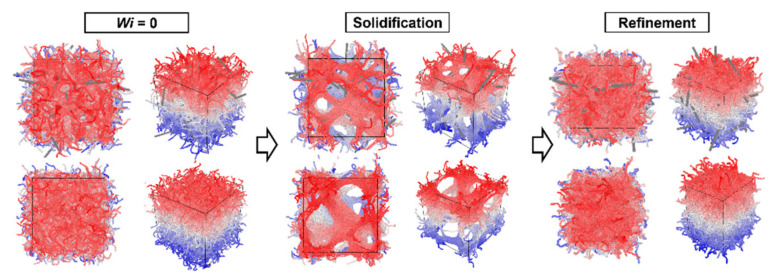
The nanostructure formation process of the PPTA system containing CNTs with Wi of 0 (three-view drawing and stereoscopic view). The direction of CNTs is disordered, and it aggregates with PPTA molecular chains to form microfibers, which become a spatial network structure.

**Figure 4 nanomaterials-12-03136-f004:**
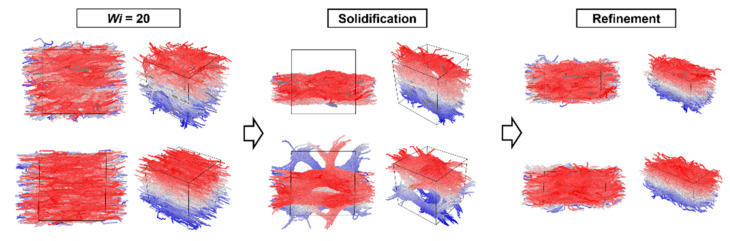
The nanostructure formation process of the PPTA system containing CNTs with Wi of 20 (three-view drawing and stereoscopic view). After shearing, both PPTA molecular chains and CNTs have obvious orientations. Then, PPTA molecular chains aggregate toward CNTs to form layered microfibers.

**Figure 5 nanomaterials-12-03136-f005:**
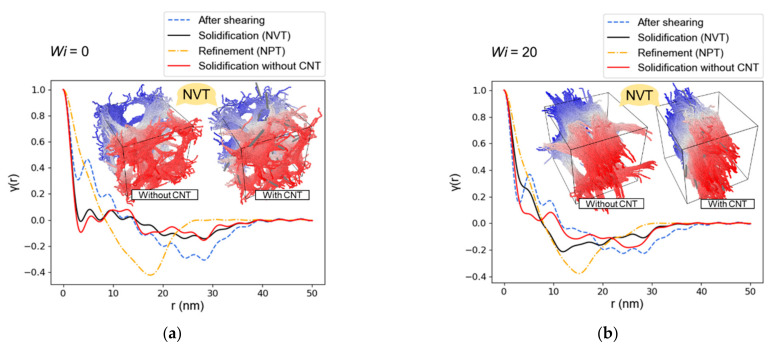
Correlation functions calculated from SAXS. (**a**) When the Wi is 0, the nanostructure of the PPTA system with CNTs is similar to that of the system without CNTs. (**b**) When the shear Wi is 20, the PPTA system with CNTs forms layered microfibers, and the correlation function of the solidification process is significantly different from that of the system without CNTs.

**Figure 6 nanomaterials-12-03136-f006:**
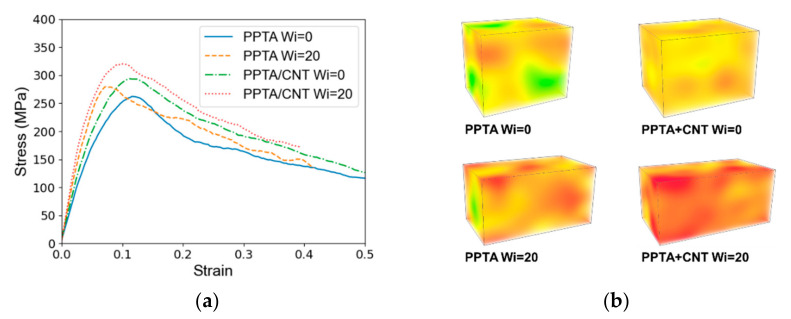
(**a**) Stress–strain curve of different fibers from Molecular dynamics simulation. (**b**) The 3D stress contour in the simulation unit of PPTA-based fibers.

**Figure 7 nanomaterials-12-03136-f007:**
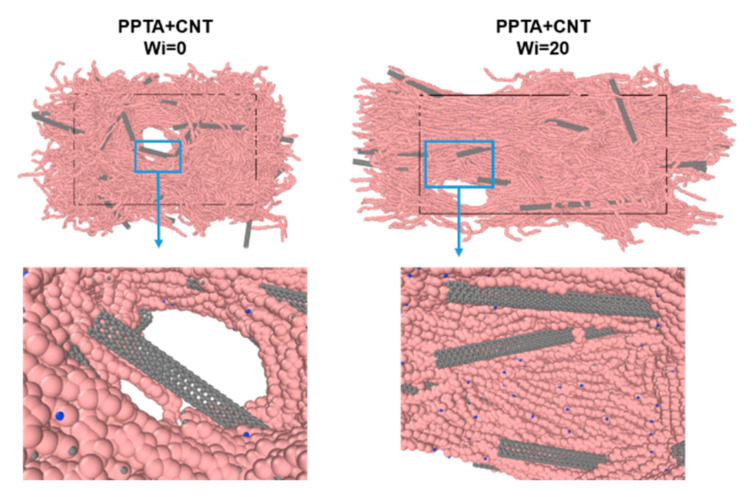
Damage initiated near the CNT reinforcements.

**Table 1 nanomaterials-12-03136-t001:** Mechanical properties of different PPTA fibers.

Materials	Shear Deformation	Strength/MPa	Young’s Modulus/GPa
PPTA	Wi = 0	262.18	3.39
	Wi = 20	279.56	4.97
PPTA + CNT	Wi = 0	293.72	3.91
	Wi = 20	320.16	5.86

## Data Availability

Not applicable.

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
