# Peer review of "Structure and Nanomechanics of PPTA-CNT Composite Fiber: A Molecular Dynamics Study"

_nanomaterials, 2022, doi:10.3390/nano12183136_

Round 1

Reviewer 1 Report

In this paper, the effect of CNT additions on the nanostructure and mechanical performance of PPTA fibers is investigated by coarse-grained molecular dynamics (CGMD) simulation. As a result, it is found that the CNT addition can improve both strength and Young’s modulus of the PPTA fiber. Such a study, which focuses on the treatment of fiber/matrix interaction is important for the development of composite products. Therefore, it is recommended that this paper is published in the Nanomaterials.

Author Response

Thank you for your encouraging comment.

Reviewer 2 Report

The paper deals with a molecular study of the mechanical behavior of PPTA-CNT composite fibres. To improve the mechanical characteristics of PPTA, the authors added CNTs to PPTA as a reinforcer. Afterwards, the new composite structure shows higher elasticity modulus and mechanical strength. This topic is interesting and timely. Actually, finding new composite structures is of great importance these days. My recommendation is to accept the paper after a minor revision based on the following comments,

* Poly (ρ-phenylene terephthalamide) (PPTA) fiber…the first parentheses is extra. It should be written as: Poly phenylene terephthalamide (PPTA) fiber.

*The descriptions about MD is insufficient. Some explanations shall be added to make clear how the MD simulation has been performed.

*The last sentence of the first paragraph of section 3.1 is unclear.

*The quality of figures 5 and 6 must be improved.

*There is no information about the distribution algorithm of CNTs inside the PPTA. Is it uniformly distributed inside the PPTA? If yes, what about other distributions?

As this reviewer knows, the distribution of CNTs while reinforcing the composite material can strongly affect the general behavior of the new composite structure.

Author Response

Summary of Revision

The authors appreciate these valuable and profound comments from the reviewers. We have carefully looked into the comments and made the corresponding amendment. Our responses to the comments are listed as follows:

Reviewer 2:

  1. Comment: Poly (ρ-phenylene terephthalamide) (PPTA) fiber…the first parentheses is extra. It should be written as: Poly phenylene terephthalamide (PPTA) fiber.

Response: These above-mentioned mistakes have been revised accordingly in the updated manuscript.

  1. Comment: The descriptions about MD is insufficient. Some explanations shall be added to make clear how the MD simulation has been performed.

Response: Modified part of Chapter 2 to explain MD simulation more clearly. (Lines 90~97)

  1. Comment: The last sentence of the first paragraph of section 3.1 is unclear.

Response: Sentences have been revised in the paper to make it more explicit. (Lines 118~121)

  1. Comment: The quality of figures 5 and 6 must be improved.

Response: The resolution of the figures in paper has been improved.

  1. There is no information about the distribution algorithm of CNTs inside the PPTA. Is it uniformly distributed inside the PPTA? If yes, what about other distributions?

Response: The insertion position of CNTs in the initial model is random in this paper, and a description has been added in the corresponding position in Chapter 2 (Line 86). Subsequent CNT distributions were simulated based on shear deformation during fiber fabrication to predict the mechanical properties of spun-produced PPTA-CNTs.
